# CFD Prediction of Ship Seakeeping and Slamming Behaviors of a Trimaran in Oblique Regular Waves

**Xiyu Liao, Zhanyang Chen \*, Hongbin Gui and Mengchao Du**

Department of Ocean Engineering, Harbin Institute of Technology at Weihai, Weihai 264209, China; 20S030035@stu.hit.edu.cn (X.L.); guihongbin@hitwh.edu.cn (H.G.); 20S030013@stu.hit.edu.cn (M.D.)
\* Correspondence: chenzhanyang@hit.edu.cn; Tel.: +86-1350-6318-766

**Abstract:** The main hull encounters waves at first and causes waves to break, when trimarans are subject to the slamming in head waves. At this moment, emergence phenomena of side hulls will not occur. Thus, the slamming study of trimarans in oblique waves presents further practical significance. In this study, a CFD method is used for trimaran seakeeping and slamming analysis. An overset grid technique is adopted to simulate ship motions in waves. Firstly, to further verify the present method, a series of verification and validation studies is conducted. Then, the motion responses and slamming pressure with different control parameters, such as forward speed and ship heading angle, are calculated and discussed. The comparative results indicate that the seakeeping and slamming behaviors of trimarans differ significantly from those of conventional monohull ships. Finally, severe bow slamming and green water in oblique waves are also observed and investigated, which should be given enough attention during ship design and evaluation.

**Keywords:** CFD; seakeeping; slamming; trimaran; oblique waves

## 1. Introduction

Recently, high-performance ships have been attracting the attention of more and more researchers. Trimarans with good maneuverability and high stability have wide development prospects [1]. When navigating in severe sea states, the phenomena of emergence and water entry of the bow and stern of ships will occur frequently, which may lead to slamming. Generally speaking, slamming is a transient process. The duration of pressure generated by transient impact is very short, but the peak value is often extremely large. The instantaneous impact of water entry will lead to local structural deformation of the hull and even structural failure. The complex cross-deck structure makes it more difficult to study the vibration of trimarans [2].

At present, the related research on trimaran slamming mainly focuses on the simulation calculation and water-entry test by using the simplified hull section model. Nikfarjam [3] presents an experimental investigation on the pressure distribution on three wedge sections with 15°, 20° and 30° deadrise during water-entry. The results give an appropriate approximation of the maximum pressures by the model resembling high-speed craft's hull sections, which can be used to estimate impact loads in different operational conditions. Zong et al. [4] conducted a vertical water entry experiment of a 2-D trimaran section from different drop heights, and flow fields were conducted using PIV technology for each condition. It was found that the flow particle velocity at the bottom of the trimaran model and the free surface between the main and side hulls was higher. By means of Flow-3D which is a commercial computational fluid dynamics code, Ghadimi et al. [5] simulated seakeeping of a wave-piercing trimaran in the presence of irregular waves via standard Bretschneider spectrum in sea state 5 in various seagoing modes. In the slamming analysis of trimaran, the relative vertical velocity of bow obtained from seakeeping analysis is used to study the problem of hull section entering the water in the most serious slamming mode. Wu et al. [6] reported the slamming loads on a typical trimaran section, based on both experimental

measurements and numerical results. The convergence of the MPS method in numerical simulation is verified by comparing the results of different particle configurations and time-step sizes. The results show that the acceleration and slamming pressure simulated by the MPS method are in good agreement with the experimental results. Sun et al. [7] analyzed the simulation of different types of motions. It was found that the characteristics of the dynamics during slamming strongly correlate to penetration depth, regardless of the types of entry.

The above research mainly considers the slamming caused by trimaran longitudinal/vertical motion in the head wave. In actual navigation, it is difficult for a ship to always keep in the head wave. Since there is an angle between the hull and the wave propagation in oblique waves, the transverse flow may cause the hull to roll, which will make the side hulls periodically shake around the main hull. When the roll angle reaches a certain value, the phenomena of emergence and water entry of the side hull will occur frequently. Differing from the slamming caused by longitudinal/vertical motion, the slamming caused by transverse motion is unique to a trimaran.

In addition, many researchers pay more attention to the hull vibration induced by waves in all directions, but there are few studies on the impact of wave loads on trimaran slamming based on the whole ship model [8,9]. Therefore, it is necessary to study the behavior of the hull slamming that occurs in waves based on ship motion. Jiao and Huang [10,11] simulated ship motions in both uni- and bi-directional regular waves based on the CFD method. Moreover, severe bow slamming and green water under bi-directional waves are also observed and investigated. Lin [12] numerically and experimentally investigated the ship motions and slamming pressures of a 10,000. TEU container ship in regular head waves. By calculating, they found the duration of the slamming, especially under parametric roll conditions, mainly depends on the rolling period and encountering wave.

The hydrodynamic response analysis of trimaran has always been the authors' research focus. In the authors' previous work [9], the nonlinear hydroelastic responses of the trimaran are analyzed by means of the segmented model experiment. The present study is the further expansion of previous studies. This study aims to conduct a comparative study on the motion and slamming responses of trimaran advancing in oblique regular waves by CFD simulations, which will give a better understanding of the influence of wave direction on trimaran slamming performance. The main novelty of this study is disclosing the motion characteristics leading to trimaran slamming as well as the relationship between slamming position and slamming time in oblique regular waves. The conclusion drawn in this study would guide the safety of ships navigating in waves.

## 2. Numerical Model Set-Up

The numerical calculations were applied to a trimaran model based on the CFD technique. Details of the CFD model and numerical method are presented in this section.

### 2.1. Ship Hull and Simulation Conditions

Only the naked hull is involved in the seakeeping and slamming investigation. The rudder, propeller or bilge keels are not appended to the model. The 3-D model of the trimaran is built by using the CFD software package STAR-CCM+, as shown in Figure 1, which gives a side and a top view of the model. The main dimensions of the ship model are listed in Table 1, where VCG denotes vertical center of gravity, BL denotes baseline.

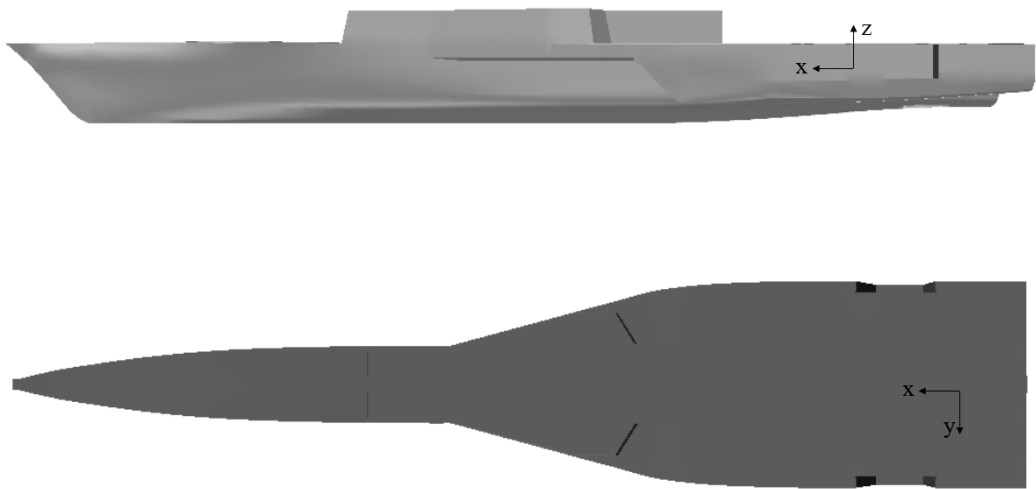

**Figure 1.** Overview of the trimaran.

**Table 1.** Main particulars of the trimaran.

| Item | Value |
|------|-------|
| Length overall $L_{OA}$/m | 142 |
| Breadth $B$/m | 27 |
| Depth $D$/m | 13 |
| Draught $T$/m | 6 |
| Displacement $\Delta$/ton | 397 |
| VCG from BL $Z_G$/m | 7.85 |
| Length of a side hull $L_P$/m | 62 |
| Breadth of a side hull $B_P$/m | 3.76 |
| Depth of a side hull $D_P$/m | 8.26 |

This study mainly focuses on investigating ship large amplitude motions and slamming pressures in oblique regular waves. Typical wave heading angles are defined in Table 2. The port side of the trimaran is the wave-forward side, and the starboard side is the wave backward side. The CFD simulation conditions in this study are listed in Table 3. The wave heading in the range of 0°–135° was selected for analysis and comparison, so the selected forward speed and wave height are relatively gentle.

**Table 2.** Definition of wave headings.

| Wave Headings | Value |
|---------------|-------|
| Head wave | 0° |
| Bow quarter wave | 45° |
| Beam wave | 90° |
| Stern quartering wave | 135° |
| Following wave | 180° |

**Table 3.** CFD simulation conditions.

| Wavelength/$\lambda$ | Wave Height/$H$ | Forward Speed/$v$ | Wave Heading/$\theta$ |
|----------------------|-----------------|-------------------|------------------------|
| 146 m | 6 m | 6.5 m/s | 0°, 15°, 30°, 45°, 135° |
|  |  | 5 m/s | 0°, 45°, 135° |

In addition, according to the Equation (1) provided by STAR CCM + UserGuide_14.04 [13], the time step $\tau$ is calculated by using wave period $P$ and number of mesh $n$,

$$\tau = \frac{P}{2.4n} \tag{1}$$

By calculation, $\tau \approx 0.0503$. Thus, in this study, the time step is 0.05 s. The wave period used in this paper is 200 times the time step, which meets the requirements recommended by ITTC [14,15].

### 2.2. Numerical Scheme

The CFD model is developed based on the Unsteady Reynolds-averaged Navier–Stokes (URANS) method. The turbulence model of Realizable $k$-$\varepsilon$ is used in this study. The Volume of Fluid (VOF) method is used to capture the free surface between air and water. Figure 2 illustrates the free surface in the established CFD model by showing the water volume fraction profile around the hull surface. As shown in the scale bar, the volume fraction of water is used to reflect the status of the computational cell. The value of 0.5 means that a computational cell is filled with 50% water and 50% air, which represents the free surface. The value of 0 and 1 represents a computational cell is filled with air and water, respectively.

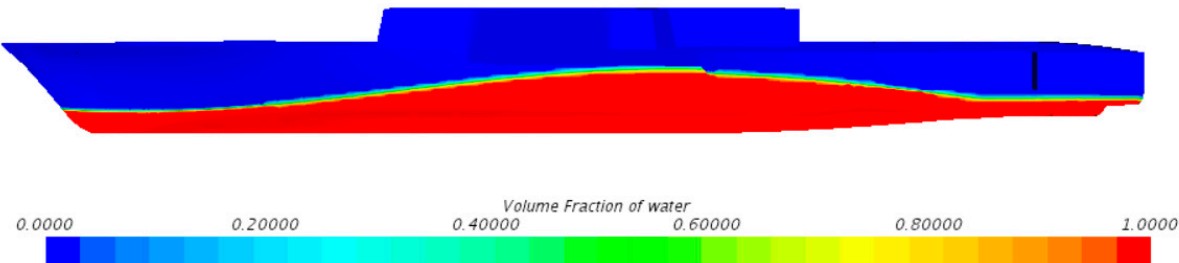

**Figure 2.** Water volume fraction profile around the hull surface.

Dynamic Fluid Body Interaction (DFBI) module is adopted to simulate trimaran motions in oblique waves. The release time is set to 0.3 s and the buffer time is set to 0.6 s. It is confirmed that this module can obtain the ship motion responses in waves conveniently and accurately [10,11,16]. However, if six Degrees of Freedom (6-DOF) of the trimaran are released, the ship will not have course stability in oblique waves. Thus, 3-DOF of the trimaran that include roll, pitch and heave are released to ensure that the trimaran does not deviate from the given route. Since the hull is fixed in the direction of $X$ and $Y$ axes, the incoming velocity can be regarded as the trimaran speed. As shown in Figure 3, according to the velocity decomposition theorem, the incoming velocity $v$ can be decomposed into $v_l$ (along the ship length) and $v_d$ (transverse flow velocity).

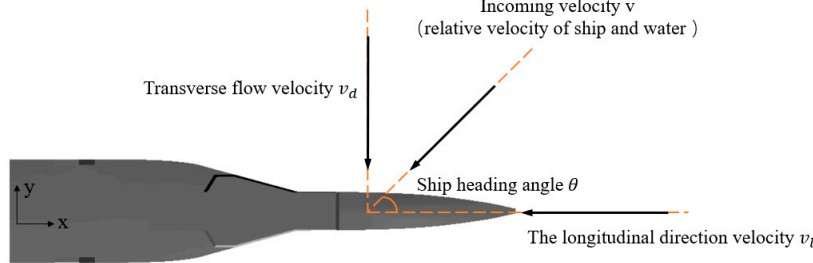

**Figure 3.** Decomposition of incoming flow velocity.

### 2.3. Fluid Domain and Boundary Conditions

An overset mesh scheme, which has been widely used by relevant scholars in simulating complex bodies with large amplitude motions [10–12,16–21], is applied to simulate the trimaran motions in waves. The computational fluid domain consists of a background region and an overset region, and the overset meshes are used for two regions to exchange data.

In this paper, the computational domain is established with a dimension of $-3.5\,L < x < 1.5\,L$ in length, $-2.3\,L < y < 2.1\,L$ in width and $-1.6\,L < z < 1\,L$ in height. Figure 4 presents the general view of the fluid domain and the definition of boundary conditions corresponding to $\theta = 45°$. The coordinate origin is set at the same height level with the calm water surface. The positive $x$-axis, $y$-axis and $z$-axis point to the ship bow, port side and the sky, respectively. The velocity inlet boundary condition is used at the upstream of the numerical towing tank. The outlet boundary condition is set to pressure outlet. The ship surface is set as the no-slip wall boundary condition. Moreover, the velocity boundary method is used to avoid the wave reflection caused by the boundary. Due to the complex wave heading in this paper, the boundary conditions of the fluid domain will change with the wave direction angle. The maximum simulation time is 80 s. The second-order convergence scheme is used in time and space. The CFL_I value is set to 0.5, and the CFL _U value is set to 1.

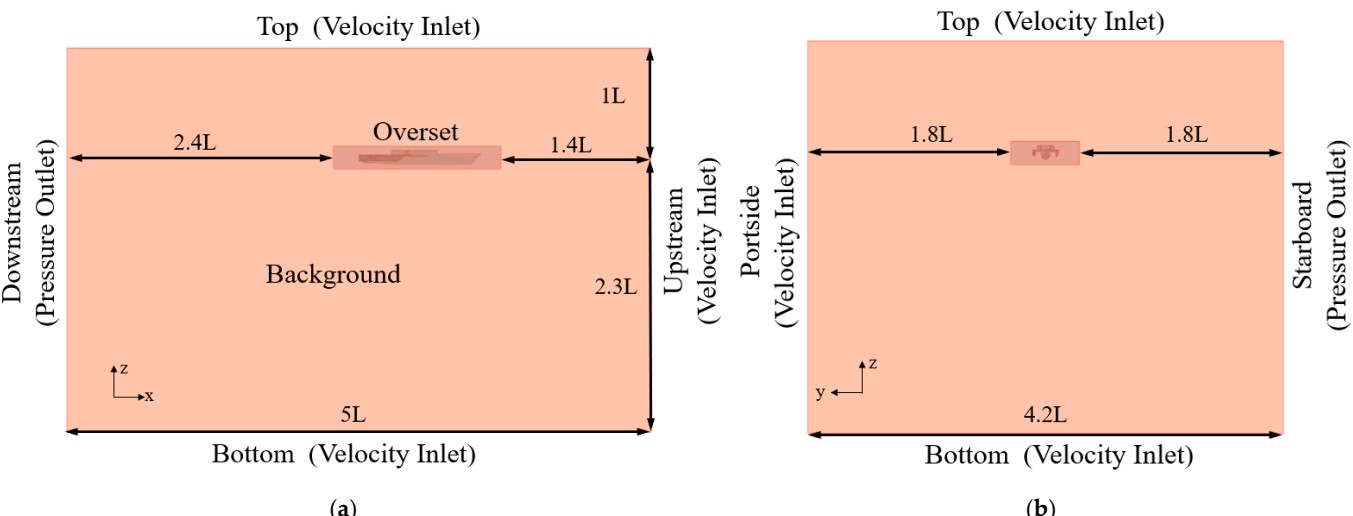

**Figure 4.** Computational domain and boundary conditions. (**a**) Side view; (**b**) Front view.

### 2.4. Monitoring Points for Slamming Pressure

In order to record the slamming pressure in oblique waves during water entry of the bow, a series of monitoring points are distributed on the main hull, side hulls and cross-decks of the trimaran, as shown in Figure 5. P1–P8 locate on the centerline of bow, and L1–L3 locate bow flare area. Moreover, to investigate the asymmetric pressure distribution on two sides of the bow in oblique waves, the monitoring points are arranged on both sides of bow symmetrically. L4–L13 locate on the left-side hull and left cross-deck, as shown in Figure 5b. (L and R represent port and starboard, respectively).

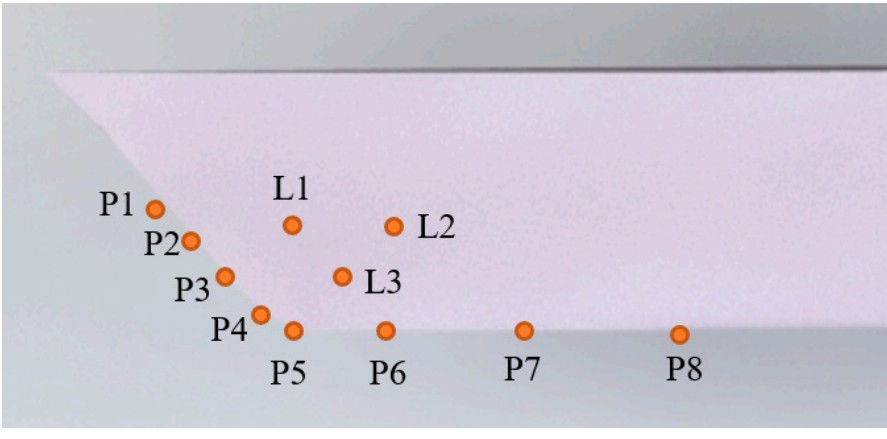

(**a**)

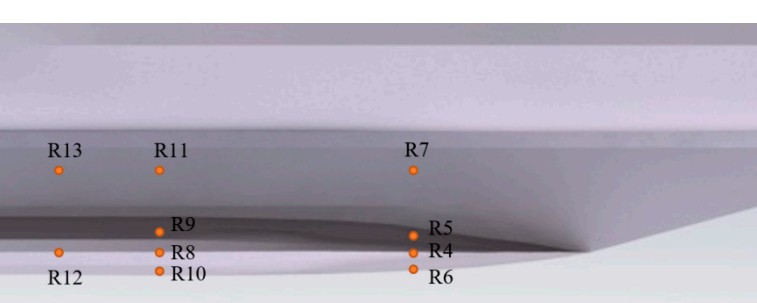

(**b**)

**Figure 5.** Monitoring points of slamming pressure. (**a**) Main hull; (**b**) Side hulls.

## 3. Verification and Validation of the Numerical Approach

In this study, the primary focus is on ship motions and slamming pressure in oblique waves. To further verify the work conducted, along with the methodology proposed above, the verification and validation studies are conducted in this section. The verification will focus on grid uncertainty analysis, and the validation will include the ability to maintain the wave elevation of oblique waves and ship motions of pitch and heave.

### 3.1. Mesh Generation

Mesh generation is performed using the automatic meshing tool of STAR-CCM+. According to Ref. [11], approximately 3.74 million cells of unstructured trimmed hexahedral mesh which include 2.7 million cells in the background region and 1.04 million in the overset region are created for numerical simulation. The general view of the computational grids is depicted in Figure 6.

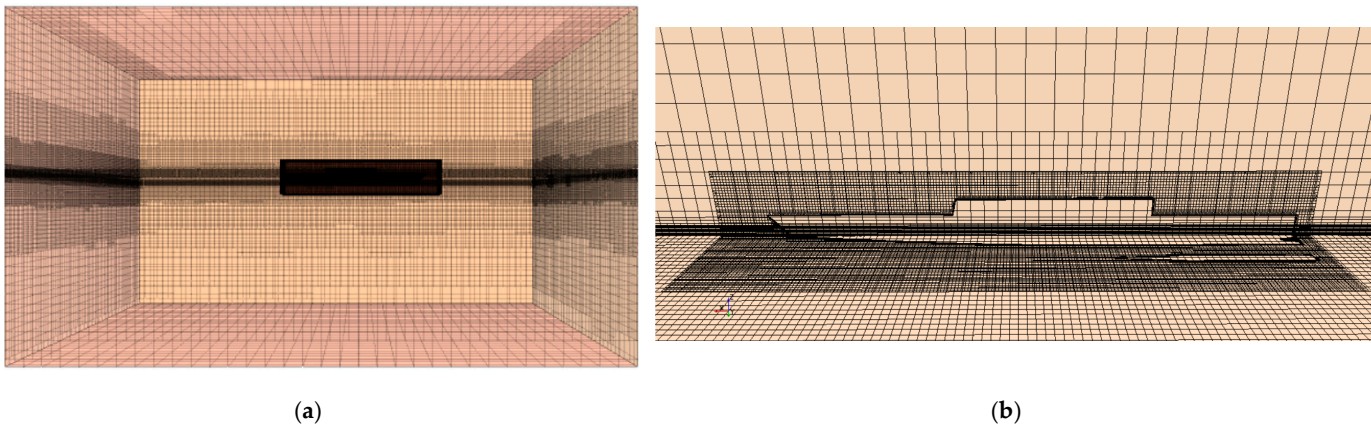

<table>
<tr><td>(a)</td><td>(b)</td></tr>
</table>

**Figure 6.** General view of the computational mesh. (**a**) Refined meshes of flow field; (**b**) Overview of the refined meshes around the hull.

The CFD mesh convergence test should be conducted first. The method of Richardson's extrapolation is used to estimate the uncertainty and numerical error [22,23]. The condition for beam sea was selected for the verification study. The condition is for wavelength $\lambda/L = 1.0$ and forward speed $v = 6.5$ m/s. The grid refinement factor $r$ is defined as:

$$r = \frac{h_{coarse}}{h_{fine}} \tag{2}$$

where $h_{coarse}$ denotes the coarser grid size, $h_{fine}$ denotes the finer grid size.

The factor $r = \sqrt{2}$ is used to divide into three different sizes of grids. Three grids named Grid 1, Grid 2 and Grid 3 are defined in this study. Grid 1 is the fine mesh grid (5.60 million cells), Grid 2 (3.74 million cells) is the medium mesh gird and Grid 3 (2.18 million cells) is the coarse mesh grid. The grid details of the hull part are shown in Figure 7.

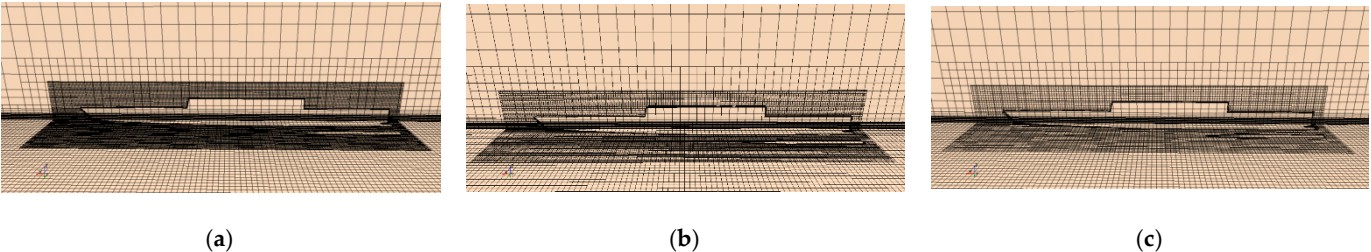

<table>
<tr><td>(a)</td><td>(b)</td><td>(c)</td></tr>
</table>

**Figure 7.** The three mesh cases for verification study (**a**) fine grid; (**b**) medium grid; (**c**) coarse grid.

The pitch motion peak and the peak pressure at M8, R4 and L4 monitoring points are selected for uncertainty analysis. Analyzed results for the grid refinement studies are summarized in Table 4. It is obvious that the relative error between Grid 1 and Grid 2 ($e^{21}$) is significantly smaller than that between Grid 2 and Grid 3 ($e^{32}$). It seems that the results of the numerical simulation tend to converge with the increased number of cells. The maximum Grid Convergence Index (GCI) is 0.03975 at monitoring point R4 and the minimum was 0.007549 at the pitch motion peak. Hence, according to Table 4, the numerical uncertainty in the fine-grid solution for the peak pressure at R4 monitoring points should be reported as 3.975%. In order to achieve a compromise between computation efficiency and accuracy, Grid 2 is used for the subsequent simulation.

**Table 4.** Grid uncertainty analysis summary.

|  | Pitch Motion Peak | M8 | R4 | L4 |
|---|---|---|---|---|
| Grid 1 | 2.988° | 67,038 Pa | 53,946.9 Pa | 31,515.8 Pa |
| Grid 2 | 3.029° | 67,693 Pa | 53,129.8 Pa | 32,187.4 Pa |
| Grid 3 | 2.895° | 68,790 Pa | 51,923.7 Pa | 30,472.9 Pa |
| $e^{21}$ | 0.01372 | 0.00977 | 0.01514 | 0.02131 |
| $e^{32}$ | 0.04432 | 0.01621 | 0.02322 | 0.05326 |
| $GCI_{fine}^{21}$ | 0.00754 | 0.01809 | 0.03975 | 0.01043 |

To capture the more accurate behavior of the free surface, The meshes around the free surface are refined. The mesh size in the $Z$ direction is 1/16 of the wave height, and the mesh size in the $X$ and $Y$ directions is 1/80 of the wave length. To reflect the slamming pressure distribution and green water more accurately, the hull surface meshes and the meshes near the pressure monitoring points are further refined. In the transition region between the overset region and the background region, a similar mesh size is used to ensure the mesh data transmission between the two regions. A boundary layer grid of 5 cells is set on the hull wall surface where wall value $y+$ is between $30-60$.

### 3.2. Numerical Towing Tank Validation

The waves generated by the CFD method above are compared with the theoretical values to verify the reliability of the numerical towing tank. Three wave probes in front of the ship are adopted for the measurement of wave elevation. The longitudinal position of probe 1 is 36 m (approximately $L/4$) from the ship's forward perpendicular. The transverse distance between probe 2 (or 3) and the ship's centerline is 30 m, as shown in Figure 8.

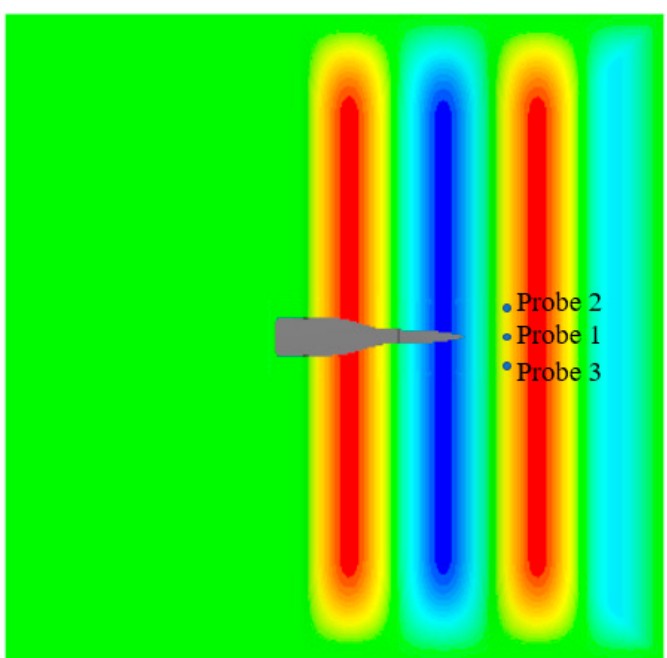

**Figure 8.** Simulated waves and wave probe arrangement.

The calm water surface is set as $z = 6$ m (wave amplitude is 3 m) for numerical simulation validation. Figure 9 shows the time histories of encountered wave elevation in one period measured at Probe 1 in comparison with the theoretical value corresponding to five different wave headings. According to the Airy wave theory, the theoretical value can be obtained by trigonometric function operations. As it is illustrated in Figure 8, all the wave amplitudes are close to 3 m in these cases, and there is no significant wave attenuation.

The maximum error is around 7.3%, which is caused by the diffraction and reflection of waves by the trimaran advancing in waves.

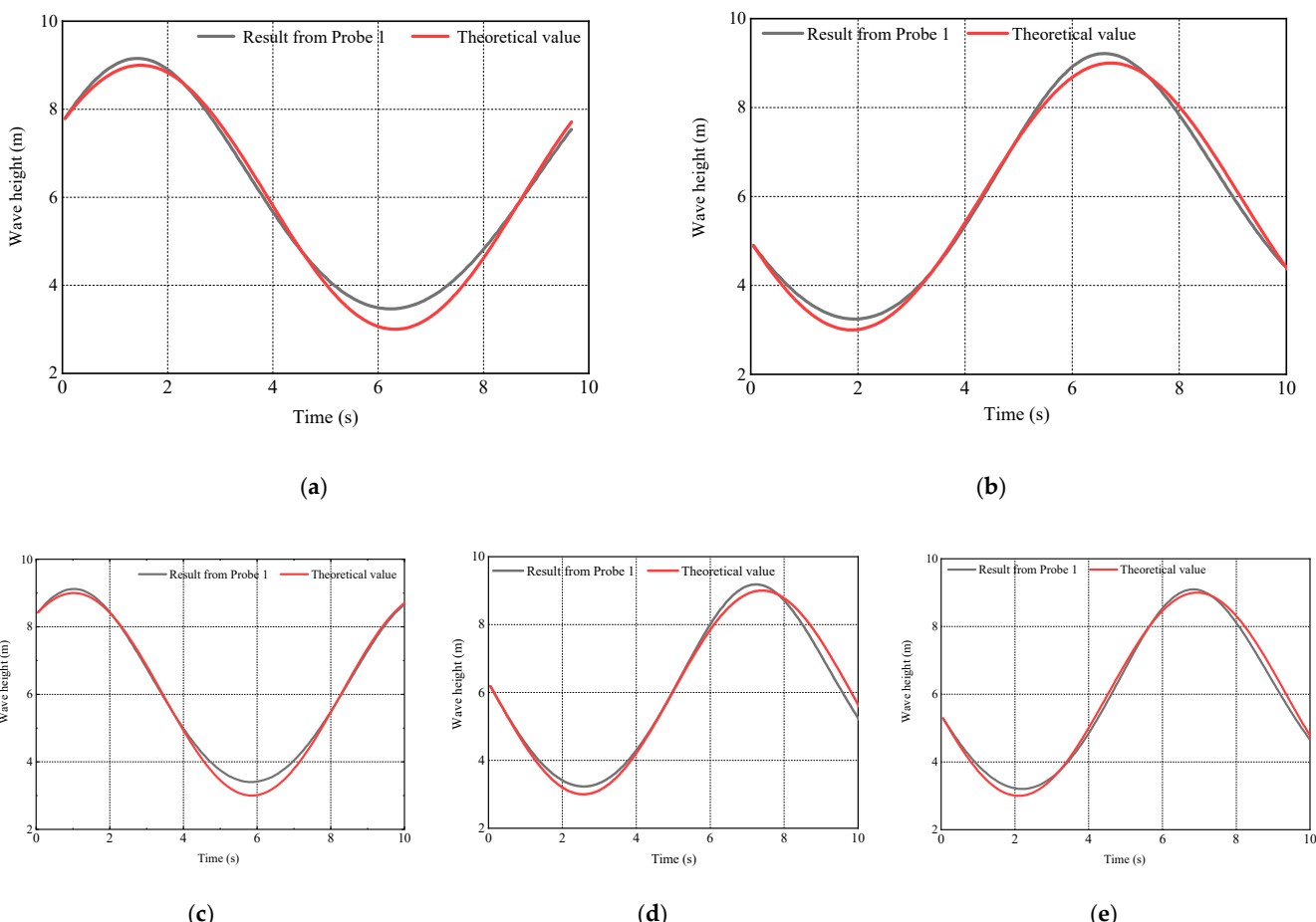

**Figure 9.** Wave elevation ($\lambda/L = 1.0$). (**a**) $\theta = 0°$; (**b**) $\theta = 15°$; (**c**) $\theta = 30°$; (**d**) $\theta = 45°$; (**e**) $\theta = 135°$.

The initialized wave profiles with the trimaran model corresponding to the five different wave headings are obtained and shown in Figure 10. It can be seen that the wave shows an elliptical shape at the outlet boundary. This is because that wave damping is set at the outlet boundary to eliminate the reflected wave. Wave damping is used at the boundary of the flow field to damp and attenuate the waves near the selected boundary, so as to reduce the oscillation near the boundary. The damping introduces vertical resistance to the vertical motion. The waves generated by the hull can be damped and attenuated to avoid the reflection of waves at the boundary. Due to the different damping of peaks and troughs, irregular elliptical boundaries appear. Moreover, the setting of damping at the outlet has little effect on the wave near the hull and will not cause the attenuation of the wave near the hull. The maximum and minimum theoretical values of wave elevation are 3 m and 6 m. The error between the maximum value and theoretical value in the figure is caused by the size difference between the grid in the overset region and the background region, which may lead to a certain deformation of the wave surface near the interface. The maximum error, which is in the head wave, reaches 2% of the wave height.

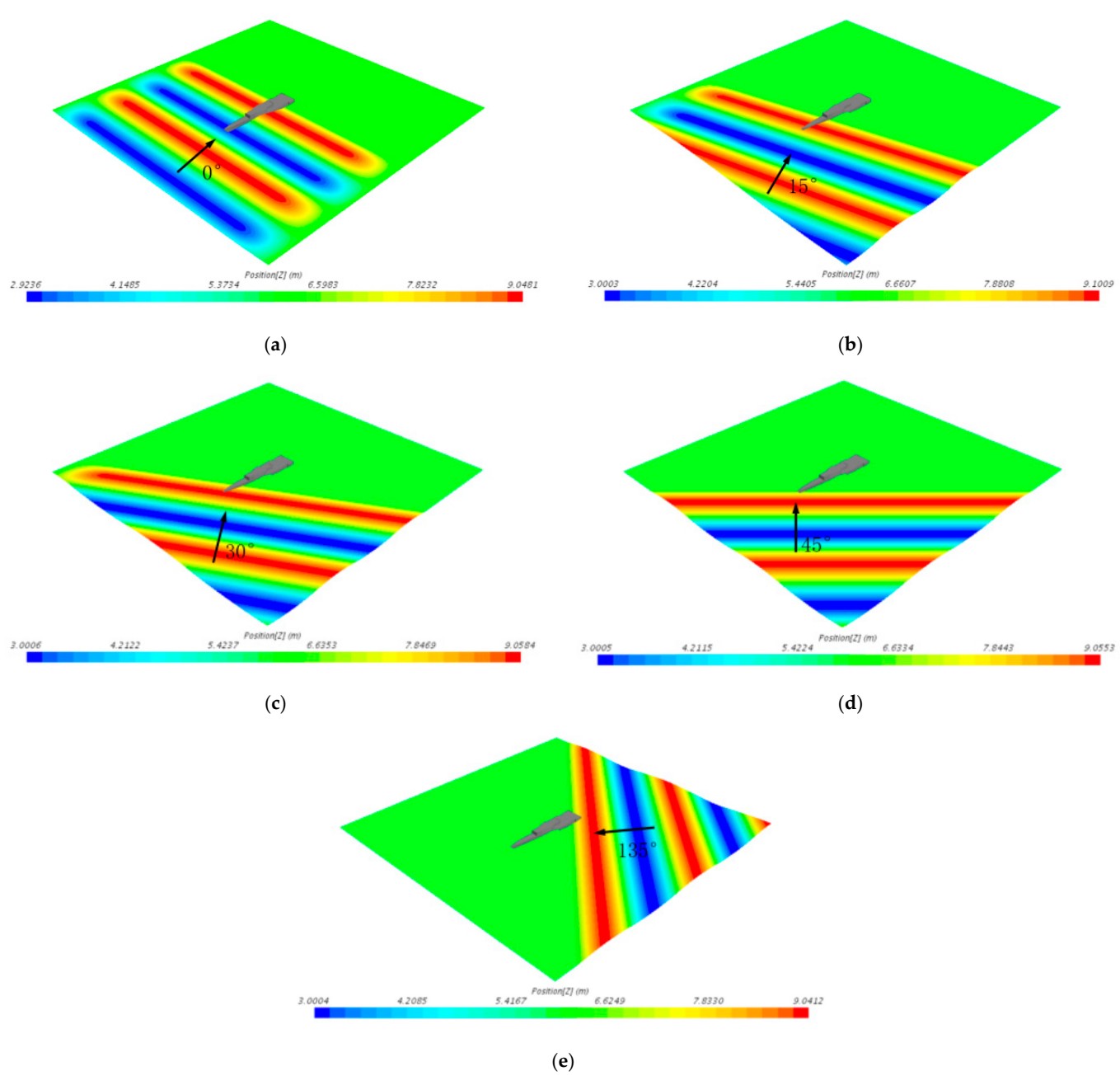

**Figure 10.** Initialized wave profile. (**a**) $\theta = 0°$; (**b**) $\theta = 15°$; (**c**) $\theta = 30°$; (**d**) $\theta = 45°$; (**e**) $\theta = 135°$.

### 3.3. Numerical Method Validation in Head Wave

Due to the lack of experimental measurement for trimaran motions in oblique waves, the present CFD method for trimaran motion prediction is validated by comparing it with the existing experimental measurement of trimaran motions in head regular waves. The dimensionless results are compared with the experimental measurement and the 3-D potential flow theoretical results by Chen et al. [9], as shown in Figure 11. The error quantification study for the head wave case is presented in Table 5.

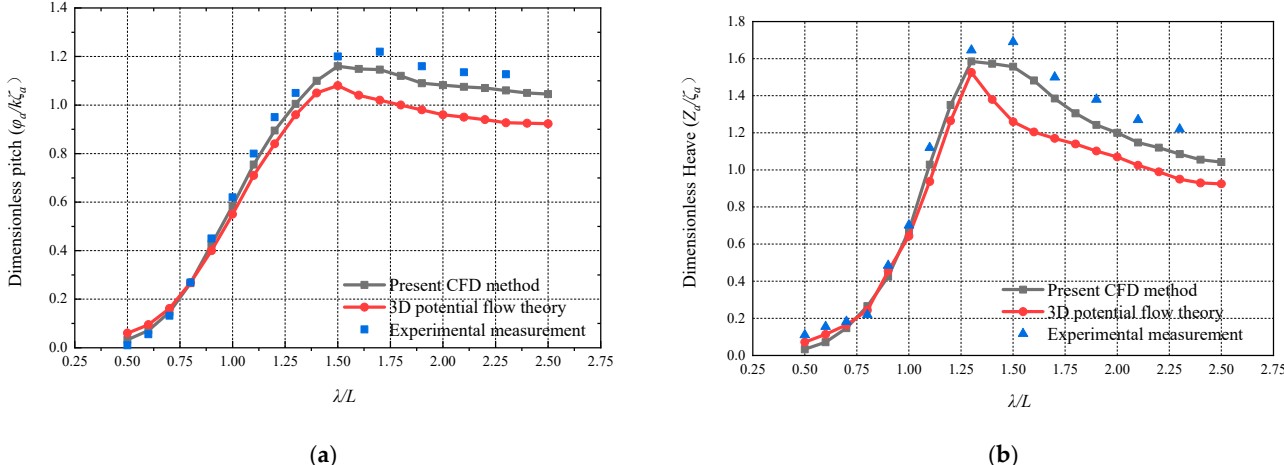

**Figure 11.** Pitch and heave responses in head wave. (**a**) Pitch; (**b**) Heave.

**Table 5.** Motion error quantification study for head wave case.

| $\lambda/L$ | Present CFD Method | | 3D Potential Flow Theory | | Experimental Data | | Error 1 | | Error 2 | |
|---|---|---|---|---|---|---|---|---|---|---|
| | Pitch | Heave | Pitch | Heave | Pitch | Heave | Pitch | Heave | Pitch | Heave |
| 0.8 | 0.266 | 0.216 | 0.268 | 0.225 | 0.269 | 0.219 | 1.12% | 1.37% | 0.37% | 2.74% |
| 0.9 | 0.425 | 0.465 | 0.4 | 0.475 | 0.45 | 0.485 | 5.56% | 4.12% | 11.11% | 2.06% |
| 1.0 | 0.585 | 0.673 | 0.55 | 0.643 | 0.62 | 0.702 | 5.65% | 4.17% | 11.29% | 8.33% |
| 1.1 | 0.755 | 1.029 | 0.71 | 0.938 | 0.8 | 1.12 | 5.62% | 8.15% | 11.25% | 16.29% |
| 1.3 | 1.005 | 1.585 | 0.96 | 1.525 | 1.05 | 1.645 | 4.29% | 3.65% | 8.57% | 7.29% |
| 1.5 | 1.16 | 1.557 | 1.08 | 1.35 | 1.2 | 1.69 | 3.33% | 7.87% | 10.00% | 20.12% |
| 1.7 | 1.146 | 1.385 | 1.02 | 1.18 | 1.22 | 1.5 | 6.07% | 7.67% | 16.39% | 21.33% |
| 1.9 | 1.09 | 1.2425 | 0.98 | 1.08 | 1.16 | 1.38 | 6.03% | 9.96% | 15.52% | 21.74% |
| 2.1 | 1.075 | 1.1475 | 0.95 | 1.025 | 1.135 | 1.27 | 5.29% | 9.65% | 16.30% | 19.29% |
| 2.3 | 1.06 | 1.085 | 0.927 | 0.95 | 1.127 | 1.22 | 5.94% | 11.07% | 17.75% | 22.13% |
| 2.5 | 1.045 | 1.0425 | 0.923 | 0.925 | 1.121 | 1.16 | 6.78% | 10.13% | 17.66% | 20.26% |

It is found from Figure 11 that the overall tendencies of the CFD calculation and 3-D potential flow theoretical results show good agreement with the experimental measurement, but the motion results obtained by the 3-D potential flow theory are smaller than those from the CFD method, which can be also found in Table 5. As shown in Table 5, error 1 denotes the differences between CFD results and experimental data, and error 2 denotes the differences between potential flow theoretical results and experimental data. Compared with the experimental measurement, the calculating accuracy of the two methods decreases as wave length range increases, but the error 2 caused by potential flow theory becomes more obvious, especially at $\lambda/L > 1.3$. This is because that the unsteady ship–wave interactions are limited to linear assumption and forward speed approximation is applied in the potential flow theory, while the CFD model can consider the nonlinear interaction between wave and ship. Therefore, the CFD numerical method can be used in the subsequent simulation of ship seakeeping and slamming behaviors.

## 4. Trimaran Motion in Oblique Regular Waves

### 4.1. Analysis of Trimaran Motions in Oblique Regular Waves

It is confirmed that the CFD numerical method adopted in this paper can simulate trimaran motion responses in head regular waves well in Section 3. Thus, the time-domain motions of the trimaran in oblique regular waves will be analyzed in this section. For the conditions of calculation time step 0.05 s, each simulation condition took about 2 days to obtain the trimaran motion and slamming simulation. In addition, the technical

specifications of the computing hardware used to conduct the numerical simulation in this paper are Intel Core i7-9700F CPU with 16 GB RAM.

### 4.1.1. Influence of Wave Heading

Figure 12 shows the time histories of motion in oblique regular waves. The results indicated that the motion signals at any wave heading present sinusoidal characteristics. Due to the different initial locations of the hull relative to the wave under the different wave headings, a phase shift will occur. Moreover, the pitch responses of trimaran in bow quarter waves, which are not much affected by wave heading, are larger than those in stern quartering waves. Unlike pitch motion, the heave motion has a close correlation with wave heading. Within the range of bow quarter waves, the heave motion increases significantly as the wave heading angle increases. Moreover, since oblique waves will cause transverse incoming flow velocity to the hull, the liquid level difference between the left and right-side hulls of the trimaran will occur. The hull pressure at the wave-forward side is significantly greater than that at the wave backward side, as shown in Figure 13.

In addition, there is an obvious phase relationship between roll and heave motions in all the wave heading cases. The roll curves always differ from the heave curves by 1/4 cycle, that is, when the roll motion peaks, the heave motion is the average value, and when the heave motion peaks, the roll motion is the average value. This indicates that there is a strong coupling relationship between roll and heave in oblique waves. Moreover, due to the side hull of the trimaran, large-amplitude roll motion may lead to a serious uneven draft of the two side hulls. At this moment, the whole ship needs to sink to a certain depth to maintain the balance of overall buoyancy and gravity, which will bring the large-amplitude heave motion.

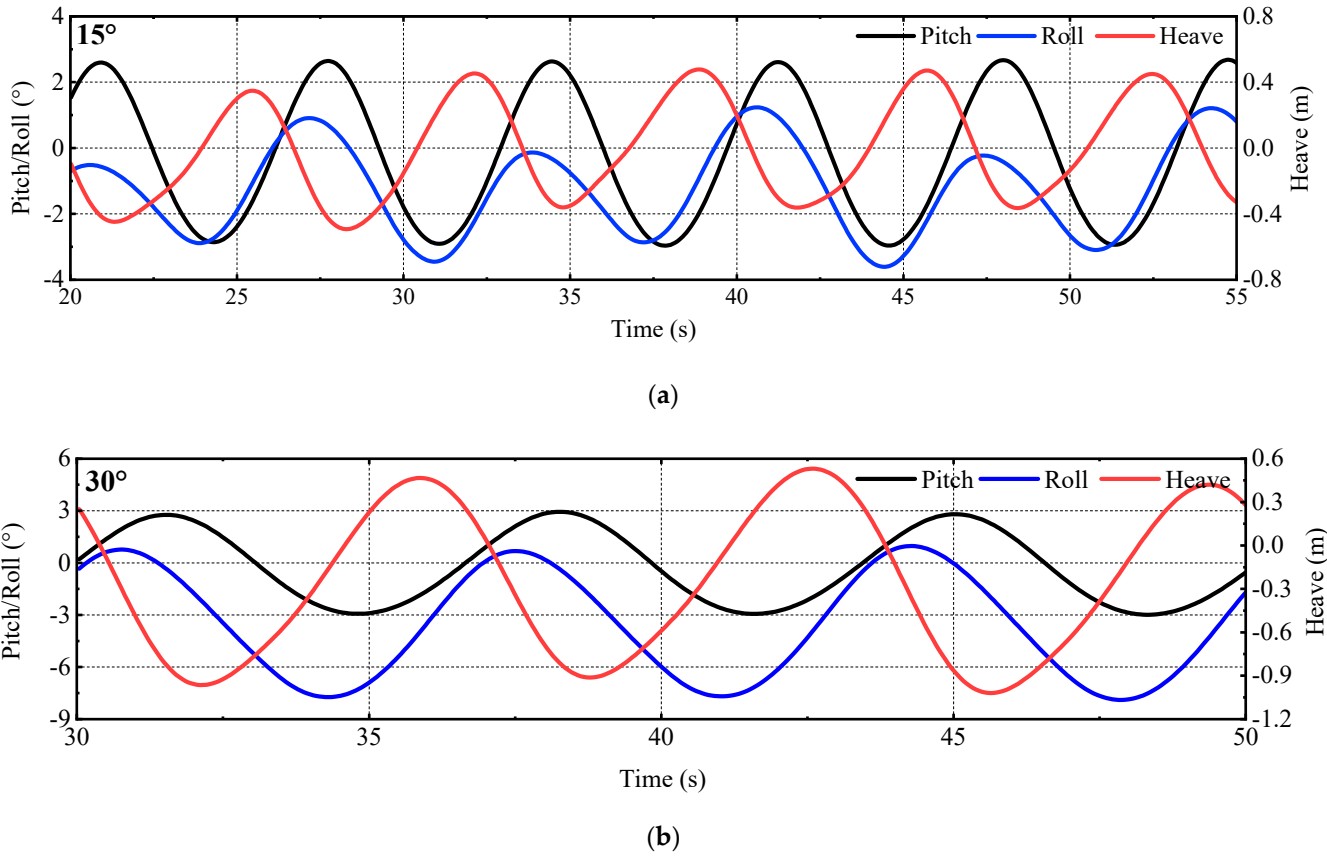

**Figure 12.** *Cont.*

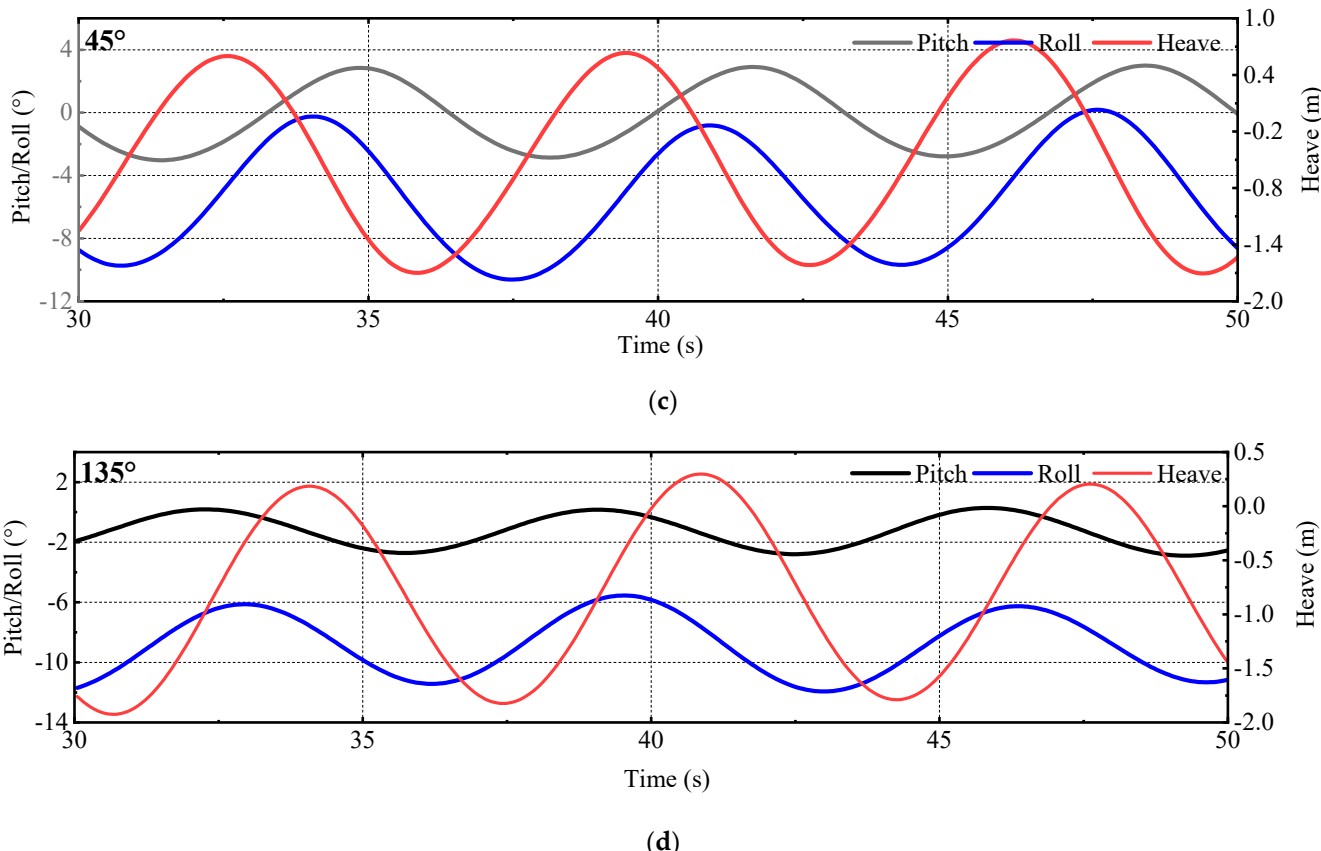

**Figure 12.** Time histories of motion in oblique regular waves. (**a**) $\theta = 15°$; (**b**) $\theta = 30°$; (**c**) $\theta = 45°$; (**d**) $\theta = 135°$.

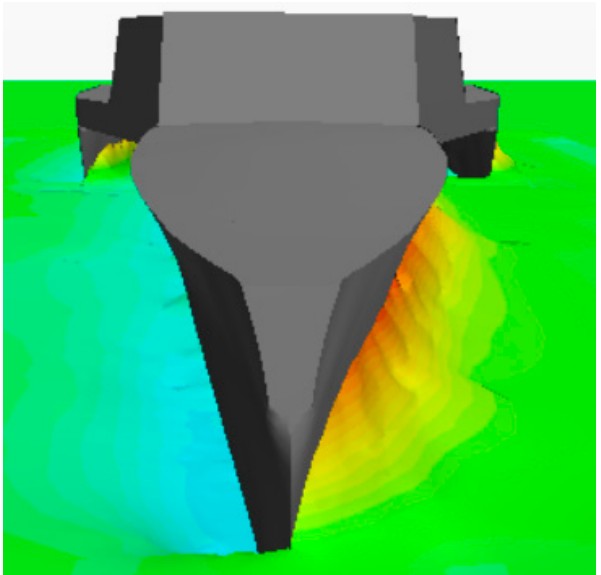

**Figure 13.** Liquid level difference between the left and right-side hulls of the trimaran in oblique waves.

### 4.1.2. Influence of Forward Speeds

To investigate the influence of forward speed on trimaran motions in oblique waves, the time-domain motion responses at $\theta = 45°$ are displayed in Figure 14. It is seen that the periods of time-domain results are different at different speeds, which are consistent with their encounter periods. The change of velocity hardly affects the pitch amplitude,

while the increase in speed will significantly increase the amplitudes of roll and heave. Moreover, the emergence phenomena of the whole left-side hull can be observed when the speed reaches 6.5 m/s. However, when the forward speed is 5 m/s, only bow emergence phenomena of left-side hull occurs, not the bottom of side hull. Thus, when advancing in the oblique waves, the trimaran should speed down to avoid the large-amplitude roll and heave motions.

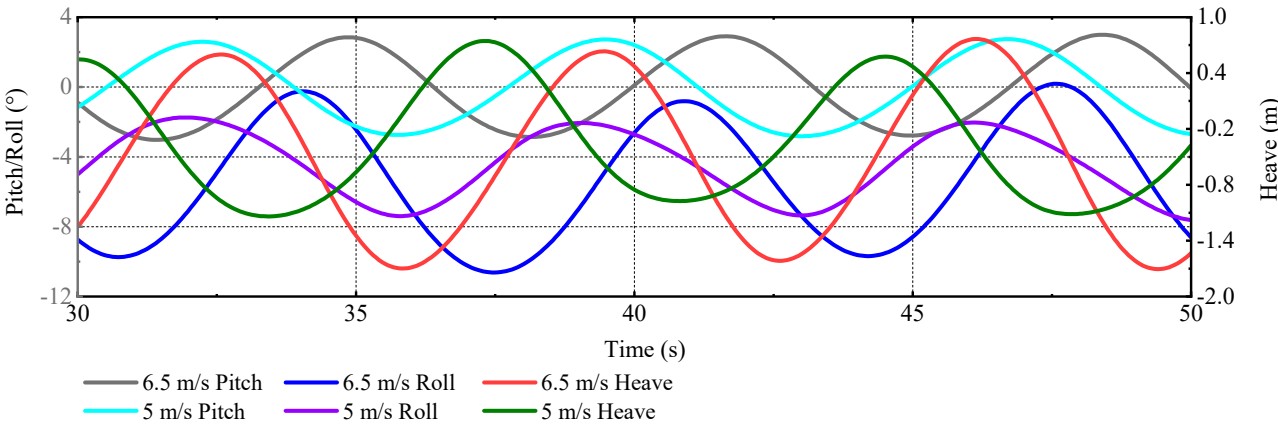

**Figure 14.** Time histories motion corresponding to different forward speeds ($\theta = 45°$).

### 4.2. Comparison of Trimaran Motions in Heading and Oblique Waves

For a better understanding of the motion behavior of trimaran in oblique waves, trimaran motion responses in heading and oblique waves are compared in this section. Differing from the head wave case, the oblique wave will lead to the transverse incoming flow. This means that one side of the hull would be subject to a continuous load, the amplitude of which changes periodically. Figure 15 presents the time histories of motions at different wave heading angles. It is obvious that the roll motion is affected by the wave direction, which is almost 0° in the head wave. With the increase in the wave heading angle, the roll amplitude increases rapidly, which is similar to the law of heave motion. The peak-to-peak value of heave motion is 2.3 m in the wave heading 45° case, which is around 2.2 times that of the head wave. Moreover, in oblique wave cases, there are a large number of phenomena of side hull emergence and green water, which will be discussed in detail in the following section ns.

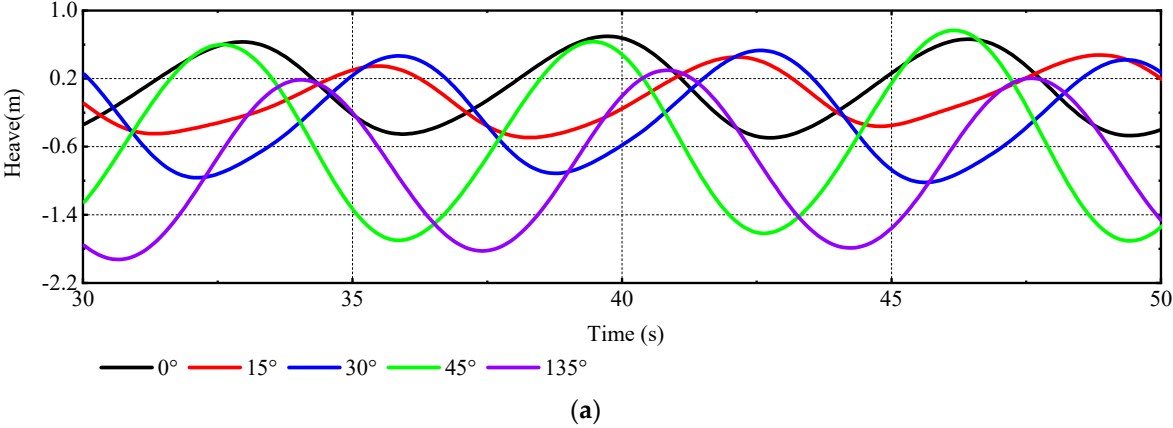

(**a**)

**Figure 15.** *Cont.*

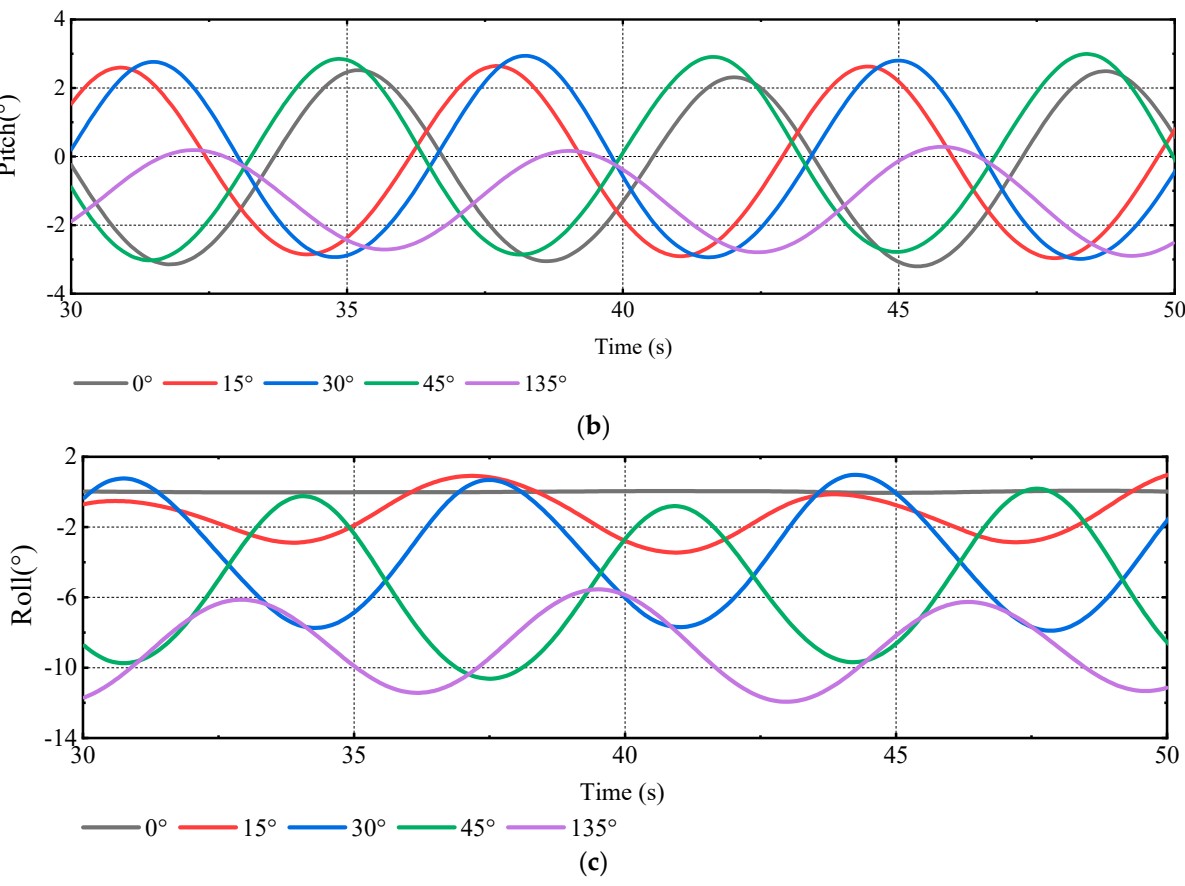

**Figure 15.** Time histories of motion at different wave heading angles (*v* = 6.5 m/s). (**a**) Heave; (**b**) Pitch; (**c**) Roll.

## 5. Slamming Analysis

It can be seen from the above analysis that there is a continuous load caused by transverse incoming flow in oblique waves, which leads to uneven pressure distribution on the left and right-side hulls of the trimaran, and even full emergence and green water of one side hull. Traditional studies on slamming of trimarans focus on numerical simulation or experiment of water entry slamming by using a wedge with simplified main hull and side hulls. However, this approach, which only considers a single period, cannot consider the actual motion state. Therefore, STAR-CCM+ is used to generate continuous waves for slamming response comparison in multi-period in this study.

### 5.1. Slamming Pressure in Oblique Waves

This section focuses on the slamming characteristics of different positions of the trimaran corresponding to the speed of 6.5 m/s. As shown in Figure 16, when advancing in oblique waves, as the port side is wave-forward side, the pressures on the port side of the main hull are greater than those on the starboard side. With the increasing wave heading angle, the uneven pressure distribution on both sides of the trimaran is gradually obvious. The wave impact to hull transfers from bow centerline area to starboard area in bow quarter waves, while the impact is mainly around the stern in stern quartering waves.

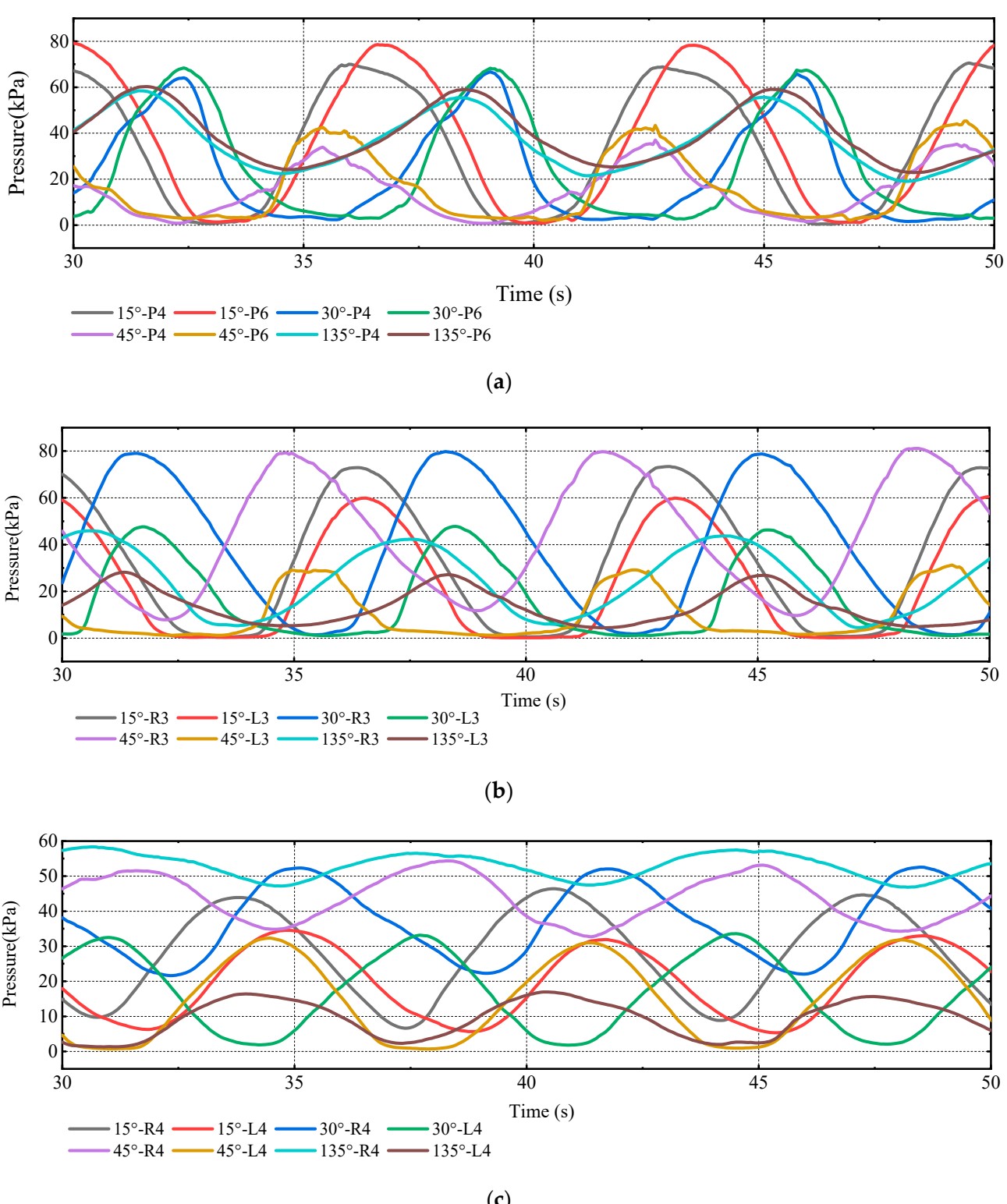

**Figure 16.** *Cont.*

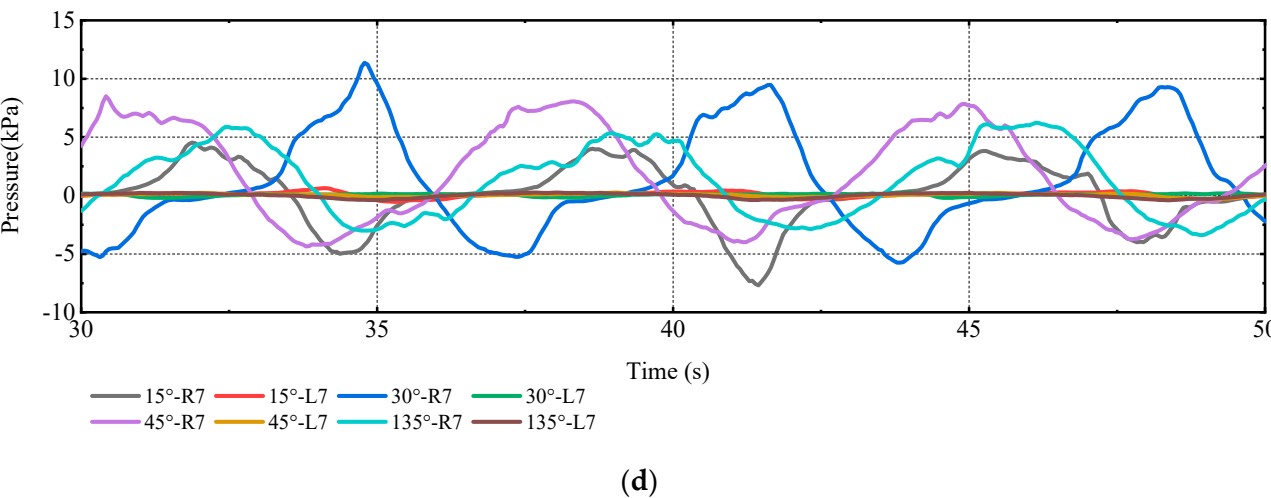

**Figure 16.** Time histories of slamming pressure at the important positions. (**a**) Centerline of the bow of main hull; (**b**) Side of main hull; (**c**) Side hull; (**d**) Cross-deck.

Due to the roll motion, the pressure distributions of the side hulls and cross-decks are becoming very complex. Firstly, there is a gradual difference in the pressure between the left and right-side hulls. With the increasing of wave heading angle, the pressure on the right-side hull decreases and the pressure on the left-side hull increases. This is because that the emergence time of the right-side hull becomes longer. Secondly, the time histories of slamming pressure on both side hulls will have a phase difference as the roll motion increases. The time difference between the two slamming peaks will increase with the increase in wave heading angle. Finally, as for the cross-decks, the water entry on the wave-forward side and splash caused by roll motion lead to a certain pressure. However, as the cross-decks are neither directly impacted by incoming flow nor immersed much below the free surface, the pressures of the cross-decks are still much smaller than those of other parts of the trimaran.

*5.2. Comparison of Slamming Pressure in Heading and Oblique Waves*

This section focuses on the difference between the trimaran slamming characteristics in head and oblique waves. The time histories of slamming pressure corresponding to different wave headings are presented in Figure 17.

Firstly, since there is little roll motion in head wave, the pressure distributions on the port and starboard sides are completely consistent. The impact caused by the incoming flow is mainly concentrated near the centerline of bow, so the pressure near the centerline is greater than that on both sides. Moreover, since the side hulls are located at the stern of the trimaran, the main hull encounters waves at first and causes waves to break in head wave. When the wave transmits to the side hulls, the energy that has decayed is not high enough to cause the side hulls and cross-decks to slam. Thus, the traditional slamming experiment for local model focuses on the head wave case, in which the weakening of the main hull to the waves along ship length in the actual navigation is ignored.

Secondly, the slamming of main hull is caused by the longitudinal/vertical motion in the head wave, while the emergence of one side hull caused by roll motion is the main cause of slamming in oblique waves. Taking the case of 45° as an example, at the same incoming flow speed, the pressure of the side hull at the wave-forward side in the case of 45° is 1.57 times that of the head wave.

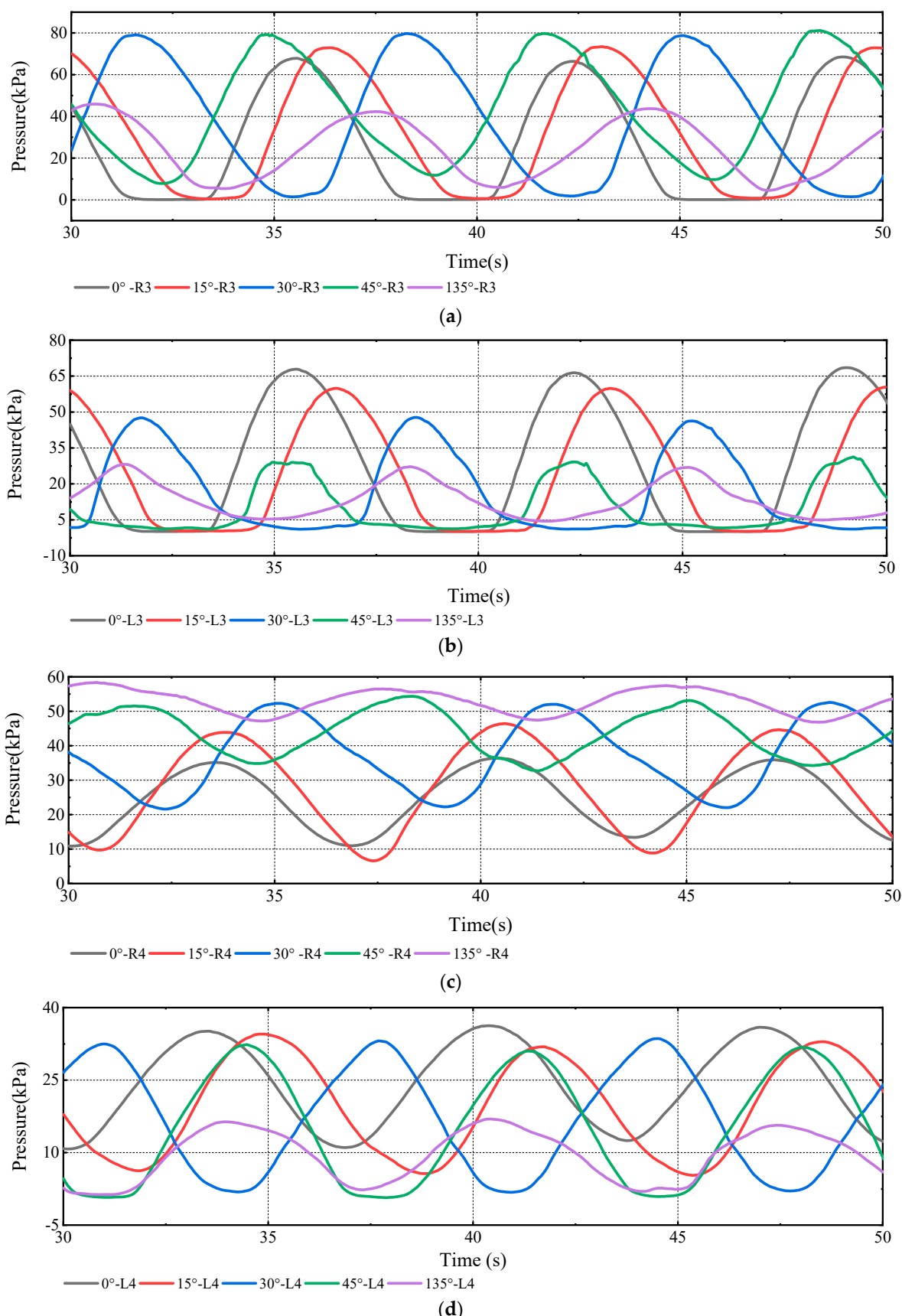

**Figure 17.** Time histories of slamming pressure corresponding to different wave headings. (**a**) Starboard side of main hull; (**b**) Port side of main hull; (**c**) Right-side hull; (**d**) Left-side hull.

Finally, the slamming of side hulls in head waves requires serious sea conditions; however, the roll motion caused by gentle oblique wave conditions may lead to the slamming phenomenon of side hulls. Compared with the main hull slamming, the slamming of side hulls and incoming flow will cause the cross-decks to bear additional bending moments. This is why the trimaran cross-decks are more likely to be damaged in waves. Therefore, compared to the head wave case with the same sea state, trimarans sail in oblique waves with greater potential safety hazards.

### 5.3. Relationship between Trimaran Motions and Slamming Pressure in Oblique Waves

The simulated impact pressures and motions at the forward perpendicular in oblique waves are presented in Figure 18.

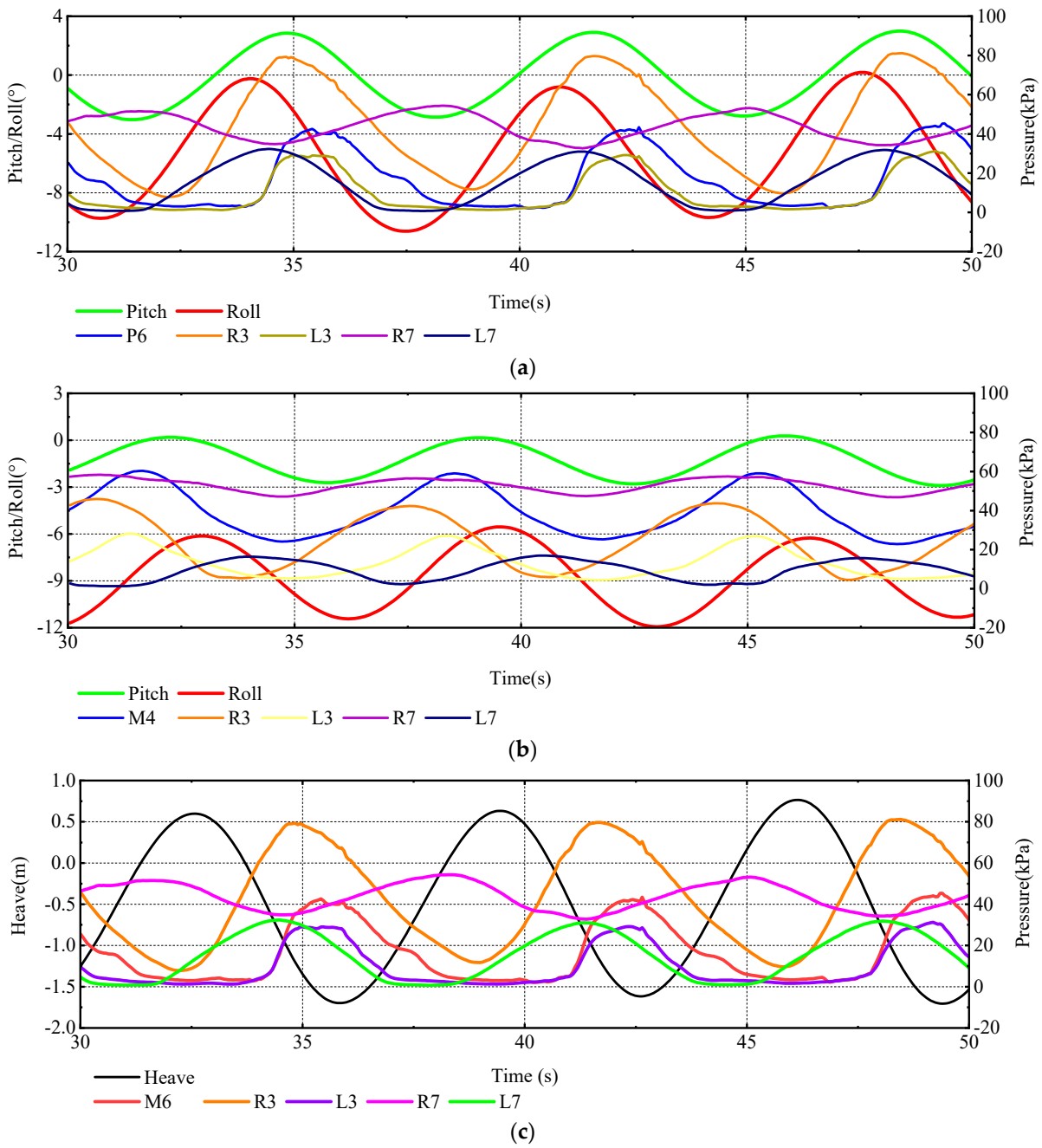

**Figure 18.** *Cont.*

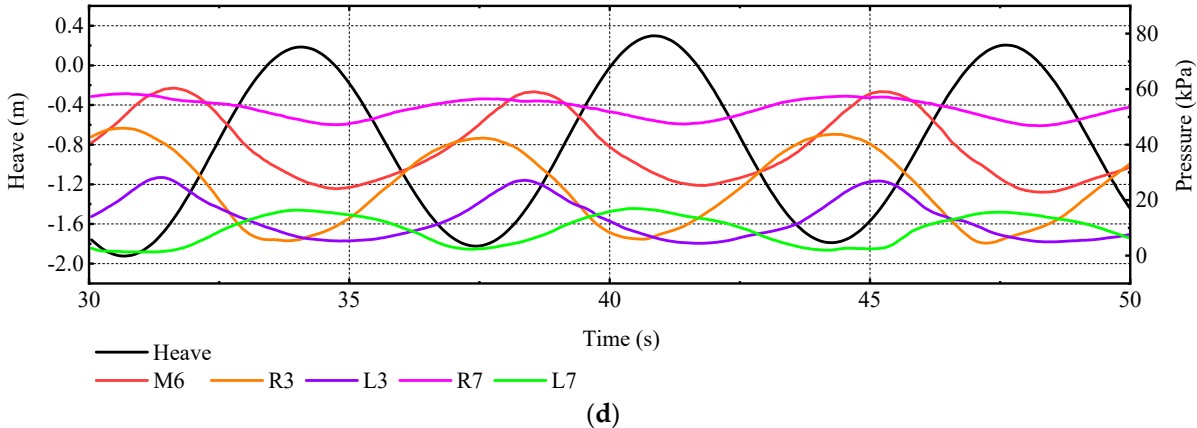

(**d**)

**Figure 18.** Time histories of motion and slamming pressure. (**a**) Roll, pitch and slamming pressure ($\theta = 45°$); (**b**) Roll, pitch and slamming pressure ($\theta = 135°$); (**c**) Heave and slamming pressure ($\theta = 45°$); (**d**) Heave and slamming pressure ($\theta = 135°$).

It is seen that the slamming pressures are generally periodical and steady during this period. The slamming pressures are closely related to trimaran heave motion, and the peak time of the two is just the opposite. The peak time of the port and starboard of the main hull is around the peak time of the bow centerline, but it does not coincide. There is no obvious relationship between the peak time of side hulls and the roll motion, which means the pressure distribution on the side hulls is not only related to the roll motion, but also affected by the heave motion.

In addition, within the range of bow quarter waves, with the increasing of wave direction angle, the amplitude of rolling motion will increase. Taking the wave heading 45° case as an example, the green water phenomenon for trimaran is depicted in Figure 19. When the bow experiences a wave crest, the green water phenomenon on the foredeck will occur, as shown in Figure 19a. Then bow experiences a wave trough, the green water on the foredeck disappears. At this moment, the stern enters the wave crest area and green water on the afterdeck occurs, as shown in Figure 19b.

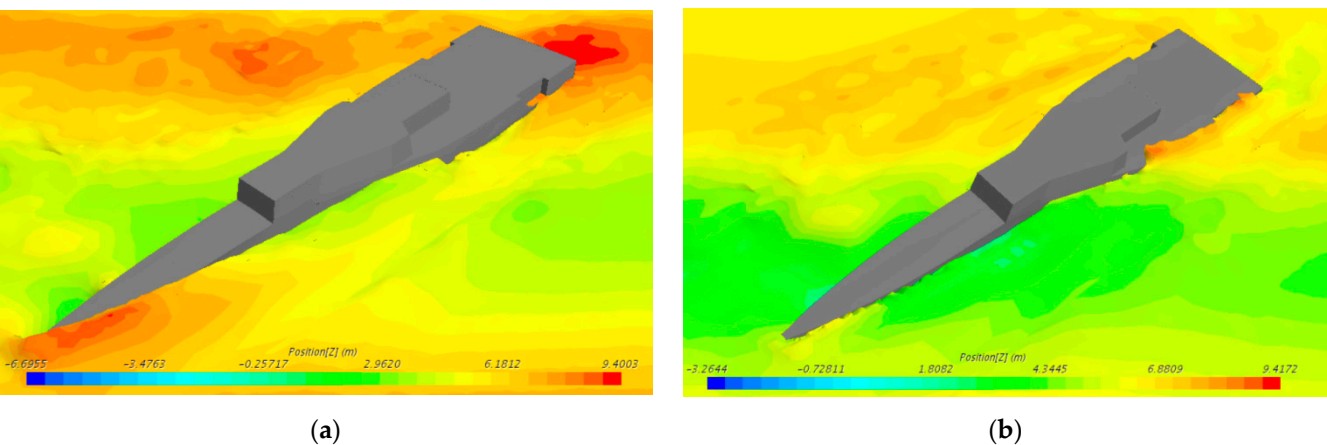

(**a**)                                   (**b**)

**Figure 19.** Phenomena of green water corresponding to $\theta = 45°$. (**a**) Green water on foredeck; (**b**) Green water on afterdeck.

Although the transverse incoming flow velocity in the wave heading 45°and 135° cases has almost the same level, the roll amplitude corresponding to $\theta = 45°$ is larger, which can be observed in figures above. This means that the trimaran is more sensitive to motion caused by wave impact on the bow. Since the area of the foredeck is much larger than that of the afterdeck, when the incoming flow velocity reaches a certain value, the amount of green water on the afterdeck caused by stern quartering waves will be significantly greater than that on the foredeck caused by bow quarter waves. As a result, the peak value of roll

motion caused by stern quartering waves is greater, which is different from monohull ships and is shown in the figures above. Although the heave amplitudes in the wave heading 45°and 135° cases have little difference, the motion time histories in stern quartering waves are below those in bow quarter waves, which means that the draft will increase when the trimaran sails in stern quartering waves.

As concluded from Section 4, due to the side hulls of the trimaran, large-amplitude roll motion may lead to the serious uneven draft of the two side hulls. Taking the wave heading 45° case as an example, when the rolling angle is up to 10°, the phenomenon of full emergence of one side hull will occur, as shown in Figure 20.

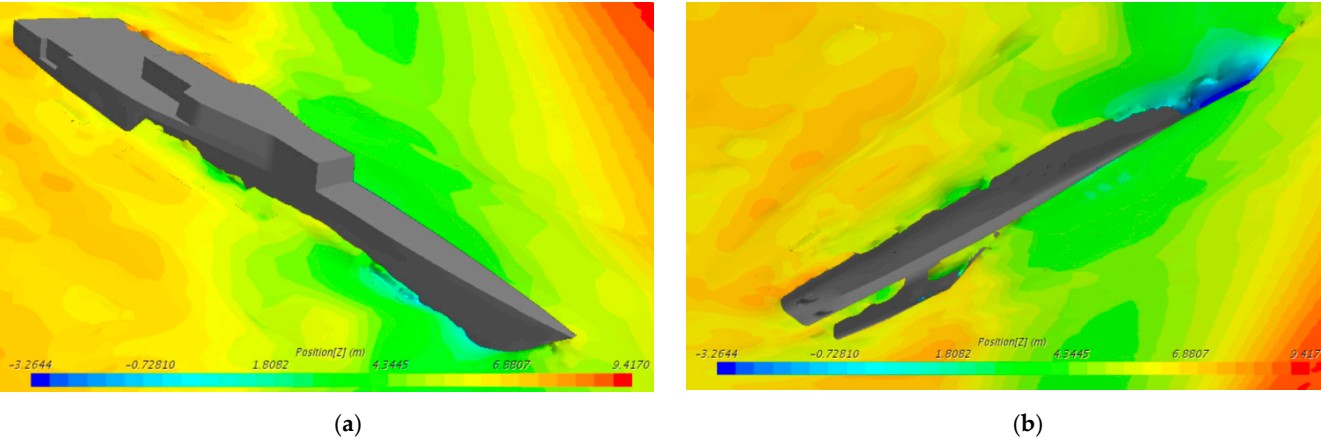

(**a**) (**b**)

**Figure 20.** Full emergence of the left-side hull. (**a**) View above the free surface; (**b**) View below the free surface.

Therefore, when advancing in stern quartering waves, the trimaran always maintains the large-amplitude roll angle and sinking range. Thus, compared to the bow quarter waves with the same angle, the trimaran sails in stern quartering waves with greater potential safety hazards.

## 6. Conclusions

This work studies the motion slamming characteristics of a trimaran in oblique regular waves based on the CFD method. The slamming and green water characteristics of the trimaran in oblique waves are also analyzed. The following conclusions are drawn according to this work:

(1) Since the side hulls are located at the stern of the trimaran, the main hull encounters waves at first and causes waves to break in head waves. When wave transmits to side hulls, the energy has decayed and the slamming of the side hulls and cross-decks is difficult to occur. Compared with the traditional slamming experiment in the head wave, the slamming study of trimarans in oblique waves presents further practical significance;

(2) When advancing in oblique waves, trimaran will have a serious roll motion that may possess a coupling relationship with the heave motion. Due to the larger mass of green water on the afterdeck, the roll peak value caused by stern quartering waves will be greater, which lead to the full emergence of one side hull and green water;

(3) The oblique waves can lead to uneven pressure distribution on the left and right-side hulls of the trimaran. At the same incoming flow speed, the pressure of side hull at wave-forward side in the case of 45° is 1.57 times of that in head wave;

(4) The differences in slamming characteristics of the trimaran in heading and oblique waves are caused by rolling motion and transverse incoming flow. When the trimaran moves in a head wave, the slamming is caused by the longitudinal/vertical motion. Due to the weakening of waves by the main hull, the side hulls and cross-decks need high enough sea state to encounter slamming. Meanwhile, in the oblique waves with

the equivalent sea state, the slamming of the side hulls caused by the joint influence of roll motion and heave motions is more obvious. Moreover, the splash and partial water entry in oblique waves lead to a certain pressure on the cross-decks;

(5)　When advancing in stern quarter waves, the trimaran always maintains the large-amplitude roll angle and sinking range. Thus, compared to the bow quarter waves with the same angle, the trimaran sails in stern quartering waves with greater potential safety hazards.

It is noted that although a detailed validation study for head wave cases is provided in this paper, an uncertainty quantification study for any one of the oblique wave cases would improve confidence in the results. Thus, seakeeping and slamming experiments of the trimaran in oblique waves are needed in the future for further validation.

**Author Contributions:** Conceptualization, X.L. and Z.C.; methodology, X.L.; investigation, Z.C.; software, X.L.; validation, Z.C.; formal analysis, M.D.; data curation, M.D.; writing—original draft preparation, X.L.; writing—review and editing, Z.C.; supervision, H.G.; project administration, H.G.; funding acquisition, Z.C. All authors have read and agreed to the published version of the manuscript.

**Funding:** This research was funded by the financial support from the Shandong Provincial Natural Science Foundation, China, grant number ZR2020ME262 and the Open Fund of State Key Laboratory of Coastal and Offshore Engineering, Dalian University of Technology, grant number LP2001.

**Institutional Review Board Statement:** Not applicable.

**Informed Consent Statement:** Not applicable.

**Data Availability Statement:** All data, models, or code generated or used during the study are available from the corresponding author by request.

**Conflicts of Interest:** The authors declared that they have no conflict of interest to this work.

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
