# Peer review of "CFD Prediction of Ship Seakeeping and Slamming Behaviors of a Trimaran in Oblique Regular Waves"

_jmse, doi:10.3390/jmse9101151_

Round 1
Reviewer 1 Report
The paper discusses pressure distribution and slamming experienced by a trimaran while encountering oblique regular waves. The study concludes that oblique waves create greater structural stress on the trimaran, particularly in stern wave cases. The paper is well written and easy to follow. The paper is of interest to people involved in multi-hull studies, especially trimarans. I recommend the paper for publication. I have just two minor comments mentioned below.
- Although the paper shows a detailed validation study for head wave cases, a verification or uncertainty quantification study is missing. An uncertainty quantification study for any one of the oblique wave cases would help improve confidence in the results.
- The paper does not mention the computational time and resources used in the study. The information should be mentioned, as it would be helpful to future readers of the paper and to people who wish to perform similar studies.
Reviewer 2 Report
paper JMSE-1384569 entitled: “CFD prediction of ship seakeeping and slamming behaviors of a trimaran in oblique regular waves”
The topic of the article is to investigate both the seakeeping responses and the slamming of trimarans in quartering seas with experimentally validated CFD numerical simulations exploiting commercial software STAR-CCM+. A brief analysis of the Numerical scheme indicating specifically the numerical parameters is included while more details concerning the physics and the numerical modeling can be found the reference manual of the commercial software as indicated. An important aspect of the study is the indication the motion characteristics that can lead to slamming and the connection between position and time of slamming for different trimaran heading angles. The authors studied thoroughly the distribution of pressure and provided times signals from many monitoring points distributed at the three hulls. In this way they reached to interesting conclusions concerning the form of slamming pressure distribution indicating also that those effects are different in trimarans and in conventional hull forms. The green water and emergence phenomena of the trimaran in oblique waves are also studied. They also investigated among others the coupling between heave and roll motion that reasonably occurs in the specific ship type and concluded to a practical conclusion cornering safety and navigation of trimarans in quartering seas.
I recommend its publication to the Journal of Marine Science and Engineering after revision according to the following directions:
1) In sec. 2.2 Numerical scheme, line 115: probably a sentence is missing after comma or just a mistyped capital letter.
2) In sec. 2.3 Fluid domain and boundary conditions, line 130: similarly.
3) In sec. 2.2 Introduction, lines 70-71: The authors have already published an interesting work in the hydroelastic responses of the trimarans. The present study that focuses of seakeeping responses and slamming of trimarans. If this is part of a more general direction of work or scientific interest of the authors, it would be really interesting and arousing and also would indicate further the practical importance of the work, to discuss present the current work as a part of more general direction somewhere at the beginning of the paper.
4) In sec. 2.2 Numerical scheme, line 115: If the dynamics of rudder and maybe other steering and positioning mechanisms like bow thrusters were modelled it should be able to calculate a stable course. In the reviewer’s opinion it should be clearly stated that the reasonable simplification of the modelling (i.e. neglection of dynamic position systems) is responsible for the problem and not the specific numerical instabilities of general possible incapabilities of the proposed method and the numerical scheme.
5) In sec. 2.3 Fluid domain and boundary conditions, line 130: In the reviewer’s opinion it should be more gentle, interesting and informative the phrase “obvious advantages” to be replaced with a few examples keeping also the reference Field (2013).
6) In sec. 2.3 Numerical wave tank validation, line 182-183: The meaning of the sentence “It can be found that the wave appears elliptical on the outlet boundary. “ is not very clear, the shape of the waveform and the deviation from the theoretical model due to absorbing layers or other techniques used should be more clearly explained.
7) In sec. 3.2 Numerical wave tank validation, line 175-176: the reviewer would suggest the phrase “theory value” to be replaced with “theoretical value”.
8) In sec. 4.1 Numerical method validation in heading wave, Fig.10 and lines 202-213: In the reviewer’s opinion the fact that inviscid solver calculations underestimate responses in comparison with viscous-solver calculations and experiments should be further justified.
9) In sec. 4.2.1 Influence of wave heading, lines 223-2225: At the beginning of the paragraph it is stated “, the pitch responses of trimaran in head quarter waves, which are not much affected by wave heading” and later at the same paragraph “Within the range of head quarter waves, the pitch motion increases significantly as the wave heading angle increases”.
10) In general, it would be more interesting from both the practical and the scientific point of view a few parameters of the problem to be given also in nondimensional form to enable generalization of the findings and connection with the physics of the underlying problem.
Reviewer 3 Report
Dear authors, please see my point of view in the attached file.

Round 2
Reviewer 2 Report
accepted
Author Response
Thank you for your comments concerning our manuscript. Those comments are all very valuable and very helpful for revising and improving our paper, as well as the important guiding significance to our future researches. We have studied comments carefully and have made corrections which we hope meet with your approval.
Reviewer 3 Report
Dear authors,
Because I had only three days to express my point of view, I will summarize my second analysis here in spite of the fact that the manuscript still has an unacceptable number of errors, which I do not want to discuss anymore. I kindly suggest yous to withdraw your manuscript, perform honestly all the numerical simulations, and compute more accurately the needed errors to make the robustness of your work credible. For more details, please see the attached document
Author Response

(The authors gave the same response as above.)
